# The Acceptability of HPV Vaccines and Perceptions of Vaccination against HPV among Physicians and Nurses in Hong Kong

**DOI:** 10.3390/ijerph16101700

**Published:** 2019-05-14

**Authors:** Teris Cheung, Joseph T.F. Lau, Johnson Z. Wang, Phoenix Mo, C.K. Siu, Rex T.H. Chan, Janice Y.S. Ho

**Affiliations:** 1School of Nursing, Hong Kong Polytechnic University, Hong Kong, China; teris.cheung@polyu.edu.hk (T.C.); chun-kwan.siu@connect.polyu.edu.hk (C.K.S.); ting-hei-rex.chan@connect.polyu.edu.hk (R.T.H.C.); 2The Jockey Club School of Public Health and Primary Care, Faculty of Medicine, The Chinese University of Hong Kong, Hong Kong, China; jlau@cuhk.edu.hk (J.T.F.L.); wangzx@cuhk.edu.hk (J.Z.W.); phoenix.mo@cuhk.edu.hk (P.M.); 3School of Public Health, Zhejiang University, Zhejiang 310058, Zhejiang Province, China

**Keywords:** HPV vaccine, acceptability, physicians, nurses, self-efficacy

## Abstract

*Background*: Human papillomavirus (HPV) is one of the most common sexually transmitted infections nationwide. *Methods*: This is the first cross-sectional survey assessing physicians’ and nurses’ knowledge of HPV and recording their attitudes to HPV vaccination in Hong Kong. Survey questions were derived from the Health Belief Model. *Results*: 1152 clinicians (170 physicians and 982 nurses) aged 21 and 60 participated in this study. A multiple stepwise regression model was used to examine associations between cognitive factors (clinicians’ attitudes) and subjects’ intention to HPV vaccine uptake. Results showed that only 30.2% of physicians and 21.2% nurses found vaccinating for HPV acceptable. *Conclusions*: Perceived self-efficacy was the only significant background and cognitive variable associated with physicians’ and nurses’ accepting HPV vaccines. Further, when nurses found HPV vaccination acceptable, cues to action was featured as a significant background variable in their choice.

## 1. Introduction

Human papillomavirus (HPV) is a well-known cause of cervical cancer and anogenital warts. HPV-16 and HPV-18 are the most common high-risk HPV forms associated with cervical cancer [1]; HPV-6 and HPV-11 are lower-risk viruses causing anogenital warts. Genital HPV infections are mainly transmitted by sexual contact (both vaginal and anal) with an infected person [2]. In order to prevent HPV infection, people can either abstain from sex or practice safer sex using condoms. They can further cut the risk of infection by having fewer sexual partners. However, epidemiologic studies suggest that up to 75% of all sexually active people will eventually become infected with HPV at some point during their lifetime [3].

In 2015, 500 new cases of cervical cancer were diagnosed at a crude incidence rate was 12.7 per 100,000 women in Hong Kong. Cervical cancer is the seventh commonest cancer among Hong Kong women. It accounts for 3.3% of all new cancer cases in females and is the ninth leading cause of all female cancer deaths. In 2016, a total of 151 women died of cervical cancer, representing 2.6% of female cancer deaths in Hong Kong [1]. Nevertheless, results from a recent Population Health Survey in 2014/15 found that the percentages of the relevant population between the ages of 25 and 64 (those ever screened and screened within the past three years) getting a cervical smear stood at 59% and 47%, respectively [4,5].

A recent systematic review (28 studies) [6] examined the acceptability of the HPV vaccination among adolescents (together with factors associated with its uptake and with adolescents declining the vaccine). Results showed the rate of vaccination across settings ranged from 2.4% to 94.4%. Scotland had the highest HPV vaccination uptake, while Hong Kong had the lowest (2.4% to 9.1%).

A recent local study asked 300 parents their views on the acceptability of HPV vaccination for their children (162 boys, 138 girls) aged 9 to 13. Results showed that the prevalence of HPV vaccination was very low (0.6% for boys against 2.2% for girls) [7].

Yuen et al. [8] conducted a local study examining the facilitators of, and barriers preventing, HPV vaccination. A total of 1229 girls aged 9 to 14 participated and were offered bivalent dose at 0 and 6 months for the period of a year. The vaccine uptake was 81.4% for the first HPV dose and 80.8% for the second. Yuen and colleagues found that two factors (the desire to prevent cervical cancer and doctors’ recommendation) were significantly associated with parents’ decision to vaccinate their daughters. Parents who had never heard of the HPV vaccine, had concerns about vaccination costs or preferred for their children to be vaccinated in a private clinic were less likely to allow their daughters to join the programme. The main reason parents refused to allow their daughters the shots was that they were worried about side effects. Another study conducted by Wang et al. [9] found that anxiety about their daughters getting cervical cancer was the most significant predictor of parental intention to get their daughters vaccinated among Chinese mothers in Hong Kong. Social influences and beliefs also affect parental intention regarding vaccination.

Past research on the acceptability of vaccination against HPV has focused on men having sex with men (SMS) [10], on female attitudes [11], parental attitudes [12,13,14,15,16,17], maternal attitudes [8,9,18,19]), and the views of baccalaureate students [20], secondary school pupils and adolescent girls [8,21,22,23]. Other research has focused on physicians’ attitudes towards prescribing HPV vaccines to patients [24,25]. Very little research examines physicians’ and nurses’ attitudes towards the acceptability of HPV vaccination or the barriers physicians have encountered in recommending HPV vaccination in the last decade [26]. Physicians and nurses are role models, whose advice is central in the promotion of good health among their patients. An understanding of clinicians’ knowledge of, and attitudes towards, HPV vaccination may begin to explain the low rate of HPV vaccine uptake in Hong Kong. The study aims to examine doctors’ and nurses’ views on the acceptability of vaccination against HPV and to determine associations between their perceptions of HPV and separately specifiable factors.

### The Health Belief Model and Outcome Measures

Extensive research has been conducted using the Health Belief Model (HBM) to examine attitudes to and beliefs about HPV and HPV vaccination. This study incorporated the HBM to explain and predict health behaviour. The HBM is a theoretical framework commonly used to guide public health interventions [27]. Its presumption is that individuals who feel susceptible to consequences in terms of their health may change their behaviour when the benefits outweigh the barriers to change or costs of adopting a new behavior [28]. Their perceived vulnerability to ill health, the perceived severity of disease (or discomfort), perceived benefit of changing, perceived barriers to change, cues to action, and self-efficacy (perceived effectiveness of their actions) are the constructs of the HBM.

## 2. Materials and Methods

### 2.1. Design, Setting and Participants

The study implemented a cross-sectional self-administrative survey using purposive sampling. Participants were recruited from public hospitals in Hong Kong. To be included, subjects needed to be male or female, licensed physicians or qualified nurses, aged between 21 and 60, and currently working full-time under the Hospital Authority (the key healthcare provider in Hong Kong). They needed to be able to read the survey language, Chinese. Part-time employees and non-readers of Chinese were excluded from the study. Participants were recruited in the hospital communal areas by research assistants who introduced the study’s aims and objectives. Respondents’ confidentiality and anonymity were assured. Participants were given a self-addressed blank envelope containing an information sheet and our survey. They were asked to return completed surveys in a sealed envelope to the research assistants on-site. Returning a completed survey implied consent. A total of 1152 eligible participants were recruited to participate in this study from June to July 2017.

### 2.2. Main Study Measures

The survey filled in by participants had 21 items and closely followed the format of Lau’s comparable study [10]. It solicited socio-demographic information including any personal history of sexually transmitted infection (STI) (both symptoms and diagnoses) and participants’ number of sexual partners over the past six months. Participants expressed views on the acceptability of vaccination on a 5-point Likert scale ranging from 0–5 (0: they found vaccination highly acceptable, or it was “very likely” they would take up vaccination, to 5: it was “very unlikely” they would take up vaccination). Four questions were asked concerning whether participants had suffered HPV-related symptoms in the past year including (1) Did you experience any burning sensation in voiding or micturition? (yes/no); (2) Did you notice any presence of white / yellowish urethral secretion? (yes/no); (3) Have you experienced any prominent growth around the genital skin or mucosa? (yes/no), and (4) Have you had any prominent lumps around the genital skin or mucosa? (yes/no). The survey also asked respondents what they knew about, and how they thought about, HPV and HPV vaccines (or how they “cognized” these, in terms of the HBM). Six composite indicator variables were constructed by counting the number of affirmative item responses reflecting how far respondents believed they were susceptible to infection by HPV (0 to 3), how severe they believed the infection was (0 to 3), whether they perceived vaccination could benefit them (0 to 5), how hard they thought it would be for get vaccinated (0 to 5), whether they thought they could vaccinate themselves easily (0 to 2) and whether they were cued to take the HPV vaccination by the media, doctors and peers (0 to 2).

### 2.3. Data Management and Statistical Analysis

The statistical analysis was carried out separately among physicians and nurses because each had a different role in HPV vaccination (i.e., prescribing as against administration). Univariate logistic regression was conducted to select significant background variables, as well as significant composite cognitive indicator variables associated with the intention to take up HPV. Odds ratios were first presented in the course of univariate analysis (OR_U_). A multiple stepwise regression model was then established using the significant univariate background variables as candidates; this allowed the calculation of multivariate odds ratios (ORm). Relationships between the cognitive indicator variables and dependent variable (respondents’ intention of taking up HPV vaccination) were then assessed, adjusting for those background variables found to be significant in the multivariate analysis; adjusted odds ratios (AOR) were also determined. Odds ratios at 95% confidence intervals (CI) were presented. Statistical analyses were performed using the statistical software SAS 9.2, with p values of 0.05 taken as statistically significant.

### 2.4. Ethics

This study was approved by the Human Subjects Ethics Sub-committee, the Institutional Review Board of a local university in Hong Kong (reference no: HSEARS20170418004).

## 3. Results

### 3.1. Socio-Demographic Background Characteristics of the Participants

A total of 1152 participants (170 physicians and 982 nurses) took part in this study. Most physician participants were male (62.9%) and most nurse participants were female (76.3%). About two-thirds (63.5%) of physicians and 54.2% of nurses were single. Over 80% of physicians (86.4%) and nurses (83.7%) were below 40 years old, with the rest of participants ranged from 41 and 60 years old. Approximately half of the participants were not religious. All the physicians had at least a bachelor’s degree, while only 56.7% of nurses had this level of qualification. Over half of the participants (57.1% physicians and 50.3% nurses) had less than five years clinical experience. Less than 30% of the physicians and nurses had more than 10 years of experience. Over 90% of physicians and nurses were frontline staff. 84.1% of physicians had a monthly income of HK$60,000 (~USD 7645) or more; however, about three quarters of the nurses (72.7%) had a monthly income of less than HK$60,000.

The doctors had on average more than 50% more sexual partners than the nurses (1.2 average for the physicians against 0.6% for the nurses). However, more nurses (19.9%) reported more than ten sexual partners than the doctors (17.6%). 6.6% of nurses and 4.2% of physicians exhibited had exhibited symptoms relating to STD in the past 6 months (Table 1).

### 3.2. Intention to Take Up HPV Vaccines (Acceptability), Knowledge on HPV Vaccines and Perceived Self-Efficacy

A total of 30.2% of physicians and 21.2% nurses reported an intention to get vaccinated against HPV (Table 2). Less than 40% of physicians (36%, *n* = 64) obtained ≥ five correct responses on knowledge of HPV vaccine compared to 44.4% of nurses (*n* = 436). 70% of physicians and 63% of nurses disagreed that they were not motivated in the HPV vaccine uptake. Rather, 37–41% of nurses and physicians respectively believed that they should take the HPV vaccination. Both physicians and nurses had a relatively high percentage of perceived self-efficacy to complete the full treatment (82% versus 85%) and a significant proportion of physicians and nurses (94% versus 97%) alleged that they were fully autonomous to take up the HPV vaccination if indicated. Physicians and nurses received peer support (76.5% versus 70.5%) and family support to take up the HPV vaccination (78.2% versus 69.3%).

### 3.3. Knowledge on HPV

The proportion giving a correct answer on individual knowledge items on HPV ranged from 50.6% to 95.3% among physicians and 51.6% to 93.6% in nurses (Table 3). Both groups had misconceptions about HPV, for instance, that the virus could not be transmitted through sex, and infection was totally curable, and that HPV 16 was able to cause genital warts. Nurses also believed that HPV could be controlled by antibiotics, that HPV was hereditary, and that HPV 18 could cause genital warts (Table 3).

Regarding perceived susceptibility, 52.4% of physicians and 49.0% of nurses saw the risk of HPV infection among women as high or very high. By contrast, 37.1% of physicians and 24.7% of nurses thought that men could easily catch the disease. 54.7% of physicians and 42.5% nurses believed that HPV was highly infectious. Yet only 9.4% of physicians and 5.7% of nurses thought they had a high or very high chance of catching HPV. 19.4% of physicians and over one-third of nurses (34.7%) were aware their knowledge of HPV was lacking. Exactly half and over half of the physicians and nurses (50.0% vs 59.8%) understood that the HPV virus could harm persons’ physical health (Table 3).

### 3.4. Cognitions on HPV Vaccine

The proportions of physicians and nurses who provided correct answers on items regarding knowledge of HPV vaccines (availability, price, the 3-shot requirement, expected duration of protection, the preferred age range for vaccination) varied, ranging from 21.2% to 93.5% in physicians and 23.6% to 79.5% in nurses. The prevalence of correct responses regarding the perceived benefits of vaccination for the physicians and nurses were similar: 71.2% (physicians) and 52.1% (nurses) knew it could prevent genital warts; 90.0% and 70.5% knew it could prevent, HPV-induced cancers (90.0% vs 70.5%); 25.9% vs 33.1% knew it could prevents STDs other than HPV; and 11.8% vs 13.8% knew it could treat genital warts. 11.8% of physicians and 14.8% of nurses knew the shots could treat HPV-induced cancers (11.8% vs 14.8%) (Table 3).

Physicians disagreed more often than nurses with statements suggesting there were serious barriers to vaccination. ‘‘HPV vaccines are expensive’’, had a 67.6% agreement rate in physicians and a 70.5% in nurses. ‘‘There are side effects’’ came in at 20.0% for physicians and 32.8% for nurses; ‘‘it is embarrassing to take up HPV vaccines’’, at 11.8% vs 18.1%, respectively; "private hospitals do not provide HPV vaccines", 10.0% vs 23.8%; "public hospitals do not provide HPV vaccination", 52.4% against 60.5%; and ‘‘getting vaccinated against HPV means you are promiscuous’’, 5.3% against 7.9%. 17.6% of doctors and 23.0% of nurses thought it was easier for healthcare professionals to be infected with HPV. 25.3% of doctors and 36.3% of nurses agreed that "healthcare professionals are less motivated to take up HPV vaccination", while also thinking (40.6% doctors, 37.1% nurses) that "healthcare professionals should take up HPV vaccination". The level of perceived self-efficacy in taking up HPV vaccination was high in physicians (82.4% and 94.1%) and nurses (85.1% and 96.7%). Both groups thought their peers (78.2% and 76.5%) and family members (69.3% and 70.5%). The rate of responses regarding cues to action among physicians and nurses were that media reports (76.5% vs 78.4%) were likely to encourage them to take up vaccination, as were doctors (52.4% vs 39.1%) and to some extent, peers (47.1% vs 46.3%). 82.9% of physicians would recommend their patients or patients’ family members to take up HPV vaccines, compared to only 59.1% of nurses (Table 3**).**

### 3.5. Factors Associated with Acceptability of HPV Vaccines

We found that gender was the only statistically significant background variable among physician participants. After adjusting for gender, the results showed four cognitive composite indicator variables as statistically significant. They were the perceived severity of disease (AOR = 2.689, 95% CI = 1.034–6.990), perceived self-efficacy in preempting it (AOR = 5.408, 95% CI = 1.466–19.945), perceived support (from family or peers) (AOR = 3.468, 95% CI = 1.191–10.099) and cues to action (AOR = 3.688, 95% CI = 1.165–11.675) (Table 4).

Adjusting for these four variables, results showed that six of the composite cognitive indicator variables were statistically significant. These were knowledge of HPV vaccines (AOR = 2.588, *p* < 0.01), perceived susceptibility to HPV (AOR = 1.779, *p* < 0.05), the perceived benefits of vaccination (AOR = 1.815, *p* < 0.05), perceived self-efficacy (AOR = 2.772, *p* < 0.01), perceived support (AOR = 2.206, *p* < 0.01) and cues to action (AOR = 2.703 to 6.960, *p* < 0.05) (Table 4). In addition, the study fitted a multiple logistic regression model containing all the significant composite cognitive indicator variables and adjusted for significant background variables. 

Four background variables were statistically significant among the nurse participants: (1) Gender (ORm = 1.764, 95% CI = 1.129–2.759), (2) age group (“21–29”: ORm = 4.095, 95% CI = 2.187–7.669; “30–39”: ORm =1.627, 95% CI = 0.802–3.301, reference group was “40 & over”), (3) having STD-related symptoms (ORm = 2.230, 95% CI = 1.779–7.446), and (4) number of sexual acts in the past 6 months (“0”: ORm = 1.854, 95% CI = 1.153–2.981; “1–5”: ORm = 2.166, 95% CI = 1.305–3.597, the reference group was “6 or more”). After fitting a multiple logistic regression model containing all the significant cognitive composite indicator variables, and adjusting for the same significant background variable, only perceived self-efficacy and cutes to action emerged as the only significant composite cognitive variables (Table 5). 

## 4. Discussion

Results of multiple logistic regression suggest that self-efficacy is the only significant composite cognitive variable associated with physicians’ and nurses’ intention get vaccinated themselves against HPV. Nurses, further, are more likely to get vaccinated when they perceive that there are positive cues to action. We found that female physicians are three times more likely than male physicians to take up HPV vaccination. This finding is unsurprising: 37.1% of respondents giving this answer were female physicians, and 86.4% (of both genders) were below 40 years old; of this group, 59.6% had been sexually active in the past 6 months. Female physicians broadly exhibited a fundamental knowledge of HPV and other sexually-transmitted diseases and cervical cancers. The main driver of their behaviour in getting vaccinated, then, is likely to be to avoid infection [29].

The acceptability of the HPV vaccine among physicians and nurses in this study is relatively low (30.2% as against 21.2%). We speculate that this low rate of acceptability may be attributed to 1) respondents having little or no exposure to STD-related symptoms themselves; 2) their having limited exposure to HPV in their clients in clinical settings; 3) their not knowing the risks of acquiring HPV transmission; 4) a misconception that HPV can be completely cured. One of the predictors for the acceptability of HPV vaccine take up is knowledge of HPV [30]. Some researchers have found that poor HPV knowledge is positively correlated with a low incidence of the intention to get vaccinated [31]. Our findings echoed Poole’s claim. It is noteworthy that only 43% of physicians and nurses answered six or more questions on HPV correctly. This ignorance among physicians and nurses in this study was unexpectedly high. Clinical specialty may also contribute to participants’ intention to take up an HPV vaccine. The possibility exists that our respondents rarely deal with young adolescents or are not confronted with patients with HPV in their clinical settings [25].

Lee and Wang [26] found that only 41.8% of healthcare providers (28 doctors and 70 nurses) would recommend HPV vaccination to their patients in sexually transmitted disease (STD) clinics in Hong Kong. It is interesting that physicians in the public hospitals were more willing to prescribe HPV vaccination than those working in STD clinics. It is encouraging to note that 82.9% of physician participants would recommend the HPV vaccination to their patients, despite rarely stating an intention to get the shots themselves. The large variation between physicians’ recommendations and individual attitudes towards the acceptability of HPV vaccination may suggest that, knowing their own behaviour, clinicians suppose themselves less at risk. It may also suggest they take pride in their efficacy as clinicians to administer all three doses.

Hong Kong is a developed part of Asia with generally high incomes. Nevertheless, there is some evidence to suggest that the general public, in both high- and low-income countries, may know little about HPV, the risk factors for cervical cancer and genital warts [32]. Decisions whether to be vaccinated for HPV are affected by individual knowledge, beliefs about vulnerability, perception of the vaccine’s effectiveness, family physicians’ views, sexual practices, cultural norms, and the cost of vaccination [33,34]. Young adolescents may rely on their peers, parents, the mass media, private general practitioners or primary healthcare providers for cues to take proactive steps to take the vaccine [20]. Of all groups, parents are most important in encouraging children to take the shots. Wang et al. [12] used telephone interviews to assess the attitudes of 1996 Chinese parents of 12 to 17-year-old unvaccinated girls towards HPV vaccination. At a 1-year follow-up, less than 10% (*n* = 97 out of 979 parents) of these participants’ daughters had received at least one dose of HPV vaccines. Wang’s findings suggest that parents’ ignorance risks their daughters’ future health.

A healthcare provider’s recommendation is one of the strongest, most consistent predictors of HPV vaccination, as evidenced by Chow et al. [33]’s survey of 480 physicians and 1,617 randomly selected mothers in Korea, Malaysia, Taiwan and Thailand. 94% of women surveyed were ignorant about HPV; 58% of women asserted that doctors should initiate the conversation about HPV vaccination. In contrast, 61% of general practitioners, 28% obstetrician-gynecologists, and 56% pediatricians claimed to be incompetent to initiate a conversation about HPV vaccination with their clients in the absence of further training. This knowledge deficit may inevitably mean that physicians fail to prescribe HPV vaccine to their clients. This only suggests the need for physicians to keep up-to-date with the science relating to STDs to reduce the risk of HPV-related diseases [35].

The study sought to determine which factors were related to a physician’s likelihood of initiating a conversation about HPV vaccination. They were likely to do so if they believed that, as professionals, they had a proactive role to play in educating women about cervical cancer and preventive health measures; if they had basic knowledge of HPV infections and vaccines; and felt comfortable raising the subject. Those less knowledgeable and with negative attitudes towards cervical cancer were less liable to begin a conversation about HPV vaccination. Some physicians may see discussions of HPV vaccination as burdensome, comparing to bringing up other vaccines offered adolescents [36]. Personal subjective bias and preferences in prescribing behaviour may lead to unnecessary delays in immunising adolescents.

It is interesting that nurses in our study suggested that ‘cues to action’ may increase their likelihood of taking up HPV vaccination. We speculate that these nurses are unlikely to come across clients suffering from STD-related diseases in their work. Their limited clinical exposure may reduce their effectiveness in caring for their own health. Tay et al. [36] surveyed 1622 nurses’ intention of getting vaccinated and examined why women might decline HPV vaccination. Results showed very few nurses (8.8%) had been vaccinated against HPV, although 12.5% planned to get vaccinated in the next 12 months. A significant proportion (44.5%) were undecided. A total of 557 (34.3%) nurses turned down HPV vaccination. The most commonly cited reasons for declining included inadequate information (49.4%), the belief that the vaccine was unproven (23.5%) and being the wrong age (25.5%). Tay’s findings [36] suggest that a professional background did not necessarily affect individuals’ perceptions of HPV vaccination as these determine their own health choices.

## 5. Limitations

We used the questions extracted by previous researchers in the HPV studies [20,29] but these questions had not been validated. There is a potential risk that some of these questions might not accurately measure of what we aimed to measure. Thus, results emerged from this study might not be generalized to similar replicated studies across countries.

## 6. Conclusions

It is evident that Hong Kong’s physicians and nurses have a very low acceptability towards HPV vaccination themselves, and that their general knowledge of HPV vaccination is barely adequate. Physicians and nurses play a pivotal role in shaping public views of HPV vaccination and correcting any misconceptions the public may have. These clinicians are often portrayed as proactive role models in infectious disease control and in preventing outbreaks in the community. The success of HPV prevention and control relies on concerted effort from stakeholders, policymakers, primary-secondary-tertiary care physicians and nurses. It is thus crucial for primary care providers to refresh their knowledge of HPV and to educate their patients about the importance of HPV vaccination in any clinical encounter.

## Figures and Tables

**Table 1 ijerph-16-01700-t001:** Frequency distributions of the background variables (*N* = 1152).

Variable	Physician (*N* = 170)		Nurse (*N* = 982)	
	N	%	N	%
Gender				
Male	107	62.9	233	23.7
Female	63	37.1	749	76.3
Age group				
21–29	99	58.2	528	53.8
30–39	48	28.2	294	29.9
40–60	23	13.5	160	16.3
Education Level				
Higher Diploma	0	0.0	285	29.0
Degree	144	84.7	557	56.7
Master or Above	26	15.3	140	14.3
Marital Status				
Unmarried	108	63.5	532	54.2
Married	62	36.5	450	45.8
Occupation				
Doctor	156	91.8	0	0.0
Psychiatrist	2	1.2	0	0.0
Houseman	12	7.1	0	0.0
RN (General)	0	0.0	797	81.2
EN (General)	0	0.0	90	9.2
RN (Psychiatric)	0	0.0	68	6.9
EN (Psychiatric)	0	0.0	14	1.4
Midwife	0	0.0	13	1.3
Income (HK$)				
20,001–40,000	8	4.7	317	32.3
40,001–60,000	19	11.2	397	40.4
60,001–80,000	60	35.3	153	15.6
80,001–100,000	32	18.8	73	7.4
>100,001	51	30.0	42	4.3
Religion				
Yes	78	45.9	477	48.6
No	92	54.1	505	51.4
Years of experience				
0–5	97	57.1	494	50.3
6–10	41	24.1	262	26.7
11–20	27	15.9	141	14.4
21–40	5	2.9	85	8.7
Main duties				
Clinical	167	98.2	901	91.8
Research	4	2.4	1	0.1
Administration	2	1.2	26	2.6
Community	0	0.0	44	4.5
STD-related symptoms				
Yes	7	4.2	64	6.6
No	159	95.8	910	93.4
Number of sex partners (past 6 months)				
0	65	40.4	327	33.7
1	94	58.4	638	65.7
≥2	2	1.2	6	0.6
Sexual behaviors (past 6 months)				
0	66	43.1	362	37.7
1–5	37	24.2	236	24.6
6–10	23	15.0	172	17.9
≥10	27	17.6	191	19.9

**Table 2 ijerph-16-01700-t002:** Perceptions related to HPV vaccines and intention to take up to HPV vaccines.

Variable	Physician (*N* = 170)		Nurse (*N* = 982)	
	N	%	N	%
Knowledge on HPV vaccines				
Availability of effective HPV vaccines				
No/Don’t know	11	6.5	201	20.5
Yes *	159	93.5	781	79.5
Availability of HPV 2/4/9 valent				
Yes *	143	84.1	691	70.4
No/Don’t know	27	15.9	291	29.6
Perceived price per shot (HK$: 1US$ = 7.8HK$)				
<800/>1500/Don’t know	86	50.6	502	51.1
800–1500 *	84	49.4	480	48.9
Number of shots required				
1–2/Don’t know	62	36.5	372	37.9
3 *	108	63.5	610	62.1
Duration of protection				
<2 year/2–5 years/Lifelong/Don’t know	134	78.8	750	76.4
6–10 years *	36	21.2	232	23.6
Age group suggested for HPV vaccination				
Above 30 years old/Don’t know	25	14.7	257	26.2
Below 30 years old *	145	85.3	725	73.8
Number of appropriate responses to the above questions on knowledge of HPV vaccines				
0	8	4.7	188	19.1
1	2	1.2	9	0.9
2	13	7.6	34	3.5
3	22	12.9	107	10.9
4	61	35.9	208	21.2
≥5	64	37.6	436	44.4
Perceived benefits of HPV Vaccines for preventing and treating diseases related to HPV				
Perceived efficacy in preventing genital warts				
No/Neutral	49	28.8	470	47.9
Yes	121	71.2	512	52.1
Perceived efficacy in preventing HPV-induced cancers				
No/Neutral	17	10.0	290	29.5
Yes	153	90.0	692	70.5
Perceived efficacy in preventing STD other than genital warts				
No/Neutral	126	74.1	657	66.9
Yes	44	25.9	325	33.1
Perceived efficacy in treating genital warts				
No/Neutral	150	88.2	846	86.2
Yes	20	11.8	136	13.8
Healthcare professionals are less motivation to take HPV vaccination				
Totally disagree/disagree	119	70.0	619	63.0
Totally agree/agree	43	25.3	356	36.3
Don’t know	8	4.7	7	0.7
Healthcare professionals should take HPV vaccination				
Totally disagree/disagree	93	54.7	609	62.0
Totally agree/agree	69	40.6	364	37.1
Don’t know	8	4.7	9	0.9
Number of item responses to the above nine questions reflecting perceived barriers related to HPV ^φ^				
0	14	8.2	81	8.2
1	45	26.5	89	9.1
2	35	20.6	223	22.7
3	35	20.6	222	22.6
4	23	13.5	162	16.5
≥5	18	10.6	205	20.9
Perceived self-efficacy to take up HPV vaccines				
I can complete the full treatment			146	14.9
Disagree	30	17.6	836	85.1
Agree	140	82.4		
I have autonomy on whether taking up HPV vaccines			32	3.3
Disagree	10	5.9	950	96.7
Agree	160	94.1		
Number of item responses to the above 2 questions reflecting perceived self-efficacy ^φ^			29	3.0
0	7	4.1	120	12.2
1	26	15.3	833	84.8
2	137	80.6		
Perceived support to take up HPV vaccines				
Family members support me to take up HPV vaccines			681	69.3
Yes	133	78.2	41	4.2
No	4	2.4	260	26.5
Don’t know	33	19.4		
Peers support me to take up HPV vaccines			692	70.5
Yes	130	76.5	33	3.4
No	7	4.1	257	26.2
Don’t know	33	19.4		
Number of item responses to the above 2 questions reflecting perceived supports ^φ^			247	25.2
0	33	19.4	97	9.9
1	11	6.5	638	65.0
2	126	74.1		
Cue to action				
Doctor recommended me to take up HPV vaccines				
Yes	89	52.4	384	39.1
No	22	12.9	182	18.5
Don’t know	59	34.7	416	42.4
Peers recommended me to take up HPV vaccines				
Yes	80	47.1	455	46.3
No	33	19.4	246	25.1
Don’t know	57	33.5	281	28.6
I have watched media reports promoting HPV vaccines				
Yes	130	76.5	770	78.4
No	13	7.6	114	11.6
Don’t know	27	15.9	98	10.0
I will recommend my patients/their family members to take up HPV vaccines				
Yes	141	82.9	580	59.1
No	29	17.1	402	40.9
Number of item responses to the above four questions reflecting cue to action received ^φ^				
0	17	10.0	119	12.1
1	19	11.2	177	18.0
2	42	24.7	244	24.8
3	31	18.2	244	24.8
4	61	35.9	198	20.2
Behavioral intention to take up HPV vaccines				
Intention to take up HPV vaccines				
Very likely	8	6.9	33	4.6
Quite likely	27	23.3	119	16.6
Not quite likely	59	50.9	431	60.1
Not likely	22	19.0	134	18.7

* Appropriate response. ^φ^ Number of affirmative responses (totally agree/agree, yes).

**Table 3 ijerph-16-01700-t003:** Frequency distributions of variables related to HPV-related perceptions (N = 1152).

Variable	Physician (*N* = 170)		Nurse (*N* = 982)	
	N	%	N	%
Knowledge on HPV				
Whether males or female could be affected by HPV				
Only Male	2	1.2	4	0.4
Only Female	4	2.4	44	4.5
Both *	162	95.3	919	93.6
Don’t know	2	1.2	15	1.5
Route of HPV transmission				
Both sexual and Mother-to-infant transmission *	86	50.6	507	51.6
Other route/Don’t know	84	49.4	475	48.4
HPV6 & HPV11 cause Genital Warts				
Yes *	142	83.5	586	59.7
No/Don’t know	28	16.5	396	40.3
HPV16 causes Genital Warts				
Yes/Don’t know	45	26.5	436	44.4
No *	122	71.8	535	54.5
HPV18 causes Genital Warts				
Yes/Don’t know	11	6.5	193	19.7
No *	159	93.5	789	80.3
HPV could be controlled by antibiotics				
Yes/Don’t know	11	6.5	193	19.7
No *	159	93.5	789	80.3
HPV could be completely cured				
Yes/Don’t know	66	38.8	419	42.7
No *	104	61.2	563	57.3
HPV is hereditary				
Yes/Don’t know	19	11.2	169	17.2
No *	151	88.8	813	82.8
Number of appropriate responses to the above questions on knowledge related to HPV ^φ^				
0	0	0.0	3	0.3
1	0	0.0	7	0.7
2	4	2.4	34	3.5
3	13	7.6	62	6.3
4	28	16.5	186	18.9
5	52	30.6	272	27.7
≥6	73	42.9	418	42.6
Perceived severity of HPV infection				
Perceived chance of contracting HPV in the future				
Low/very low	79	46.5	508	51.7
Moderate	75	44.1	418	42.6
Very high/high	16	9.4	56	5.7
Perceived damages of HPV infection on physical health				
Low/very low	9	5.3	36	3.7
Moderate	76	44.7	359	36.6
Very high/high	85	50.0	587	59.8
Perceived infectivity of HPV				
Low/very low	0	0.0	112	11.4
Moderate	77	45.3	453	46.1
Very high/high	93	54.7	417	42.5
Number of appropriate responses to the above questions reflecting perceived severity ^φ^				
0	50	29.4	332	33.8
1	55	32.4	266	27.1
≥2	65	38.2	384	39.1
Perceived susceptibility of HPV infection				
Perceived knowledge on HPV				
Low/very low	33	19.4	341	34.7
Moderate	110	64.7	577	58.8
Very high/high	27	15.9	64	6.5
Perceived prevalence of HPV infection among male				
Low/very low	18	10.6	175	17.8
Moderate	89	52.4	564	57.4
Very high/high	63	37.1	243	24.7
Perceived prevalence of HPV infection among female				
Low/very low	8	4.7	37	3.8
Moderate	73	42.9	464	47.3
Very high/high	89	52.4	481	49.0
Number of appropriate responses to the above questions reflecting perceived susceptibility ^φ^				
0	72	42.4	460	46.8
1	33	19.4	285	29.0
≥2	65	38.2	237	24.1

* Appropriate response. ^φ^ Number of affirmative responses (very high/high).

**Table 4 ijerph-16-01700-t004:** Associations between composite cognitive indicator variables and intention to take up HPV vaccines.

Physician ^a^	Row %	ORU (95%CI)	AOR (95%CI)	
Perceived severity of HPV (number of items with affirmative responses)				
0	17.1	1	1	
≥1	37.3	2.894 (1.132, 7.396) *	2.689 (1.034, 6.990) *	
Perceived self-efficacy to take up HPV (number of items with affirmative responses)				
0 or 1	10.3	1	1	
2	36.8	5.042 (1.413, 17.992) *	5.408 (1.466, 19.945) *	
Perceived support to take up HPV (number of items with affirmative responses)				
0 or 1	14.3	1	1	
2	37.0	3.529 (1.237, 10.072) *	3.468 (1.191, 10.099) *	
Cue to action to take up HPV (number of items with affirmative responses)				
0 or 1	12.5	1	1	
≥2	36.9	4.093 (1.313, 12.767) *	3.688 (1.165, 11.675) *	
***Nurse ^b^***				
Perceived susceptibility of HPV (number of items with affirmative responses)				
0	18.4	1	1	
1	21.8	1.242 (0.808, 1.908)	1.204 (0.765, 1.896)	
≥2	27.1	1.653 (1.062, 2.573) *	1.779 (1.109, 2.853) *	
Knowledge on HPV vaccines (number of items with correct answer)				
0	12.5	1	1	
1 or 2	7.5	0.568 (0.160, 2.014)	0.656 (0.179, 2.404)	
3	19.5	1.700 (0.838, 3.449)	2.013 (0.958, 4.274)	
≥4	26.5	2.465 (1.472, 4.129) ***	2.588 (1.515, 4.421) **	
Perceived benefits to take up HPV (number of items with affirmative responses)				
0	16.8	1	1	
1	14.2	0.822 (0.463, 1.461)	0.860 (0.462, 1.600)	
2	20.0	1.241 (0.691, 2.229)	1.011 (0.544, 1.880)	
≥3	31.1	2.241 (1.367, 3.676) **	1.815 (1.076, 3.063) *	
Perceived self-efficacy to take up HPV (number of items with affirmative responses)				
0 or 1	9.7	1	1	
2	23.6	2.884 (1.544, 5.387) ***	2.772 (1.422, 5.401) **	
Perceived support to take up HPV (number of items with affirmative responses)				
0	12.7	1	1	
1	24.1	2.181 (1.133, 4.200) *	1.915 (0.965, 3.798)	
2	24.9	2.289 (1.446, 3.625) ***	2.206 (1.360, 3.576) **	
Cue to action to take up HPV (number of items with affirmative responses)				
0	5.5	1	1	
1	13.0	2.575 (1.004, 6.607) *	2.703 (1.032, 7.082) *	
≥2	28.0	6.664 (2.852, 15.570) ***	6.960 (2.918, 16.597) ***	

* *p* < 0.05; ** *p* < 0.01; *** *p* < 0.001. ^a^ AOR: adjusted OR, odds ratios after adjusting gender, which was the significant background variable. ^b^ AOR: adjusted OR, odds ratios adjusted for all multivariate significant background variables, including gender, age group, STD-related symptoms and number of sexual behaviors in the past 6 months. OR_U_: univariate odds ratios. Univariate non-significant variables, not considered in the model. 95% CI: 95% confidence interval.

**Table 5 ijerph-16-01700-t005:** Multiple logistic regression result—associations between composite cognitive indicator variables and intention to take up HPV vaccines.

Variable	β	SE	AOR (95%CI)
Physician ^a^			
Perceived severity of HPV (number of items with affirmative responses)			
0	Reference		1
≥1	0.989	0.535	2.689 (0.943, 7.667) ^+^
Perceived self-efficacy of HPV (number of items with affirmative responses)			
0 or 1	Reference		1
2	1.5182	0.736	4.564 (1.079, 19.311) *
Perceived support to take up HPV (number of items with affirmative responses)			
0 or 1	Reference		1
2	1.1662	0.630	3.210 (0.934, 11.026) ^+^
Cue to action to take up HPV (number of items with affirmative responses)			
0 or 1	Reference		1
≥2	0.0771	0.713	1.080 (0.267, 4.373)
Nurse ^b^			
Perceived susceptibility of HPV (number of items with affirmative responses)			
0	Reference		1
1	0.374	0.250	1.453 (0.89, 2.373)
≥2	0.479	0.271	1.615 (0.95, 2.746) ^+^
Knowledge on HPV vaccines (number of items with appropriate answer)			
0	Reference		1
1	−0.529	0.695	0.589 (0.151, 2.300)
2	0.407	0.413	1.502 (0.669, 3.371)
3	0.569	0.298	1.767 (0.986, 3.166) ^+^
Perceived benefits to take up HPV (number of items with affirmative responses)			
0	Reference		1
1	−0.480	0.346	0.619 (0.314, 1.219)
2	−0.242	0.336	0.785 (0.407, 1.515)
≥3	0.309	0.291	1.362 (0.77, 2.412)
Perceived self-efficacy of HPV (number of items with affirmative responses)			
0 or 1	Reference		1
≥2	0.902	0.374	2.464 (1.185, 5.124) *
Perceived support to take up HPV (number of items with affirmative responses)			
0	Reference		1
1	−0.114	0.390	0.892 (0.415, 1.918)
2	0.151	0.287	1.163 (0.663, 2.041)
Cue to action to take up HPV (number of items with affirmative responses)			
0	Reference		1
1	0.699	0.517	2.012 (0.73, 5.546)
≥2	1.432	0.492	4.185 (1.595, 10.98) **

* *p* < 0.05; ** *p* < 0.01. *** *p* < 0.001. ^+^ 0.05 < *p* < 0.1; ^a^ AOR: adjusted OR, odds ratios after adjusting gender and simultaneously for all involved variables. STD-related symptoms and number ^b^ AOR: adjusted OR, odds ratios after adjusting simultaneously for all involved variables and the significant background variables which include gender, age group, number of sexual behaviors in the past 6 months. 95%CI: 95% confidence interval. SE: Standard error.

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
