# Peer review of "The Acceptability of HPV Vaccines and Perceptions of Vaccination against HPV among Physicians and Nurses in Hong Kong"

_ijerph, 2019, doi:10.3390/ijerph16101700_

Round 1

Reviewer 1 Report

The study Teris Cheung et al. finds the perception of HPV physicians and nurses and the cognitive variables related to vaccine acceptance through 21 questions taken from the Health Belief Model used to guide public health interventions. The methodology is the one used by the same authors in previous HPV publications, always with the same objectives.

A methodology that uses as a tool only the administration of questions without a validation process, risks not accurately measuring what you want to measure and the results cannot be generalized. This limit should be   reported and discussed  in the “discussion” section..

Despite this limit, the results are still important to guide future research on health professionals 'attitudes towards the HPV vaccine attitude related to patients' awareness of risk behaviors for HPV and the importance of vaccination.

Author Response

The study Teris Cheung et al. finds the perception of HPV physicians and nurses and the cognitive variables related to vaccine acceptance through 21 questions taken from the Health Belief Model used to guide public health interventions. The methodology is the one used by the same authors in previous HPV publications, always with the same objectives.

A methodology that uses as a tool only the administration of questions without a validation process, risks not accurately measuring what you want to measure and the results cannot be generalized. This limit should be   reported and discussed  in the “discussion” section..

Despite this limit, the results are still important to guide future research on health professionals 'attitudes towards the HPV vaccine attitude related to patients' awareness of risk behaviors for HPV and the importance of vaccination.

Authors’ response:

A limitation section was added in the manuscript on p. 19 as written below:

3.      Limitations

We used the questions extracted by previous researchers in the HPV studies [17, 30] but these questions had not been validated. There is a potential risk that some of these questions might not accurately measure of what we aimed to measure. Thus, results emerged from this study might not be generalized to similar replicated studies across countries.

Thank you very much for your valuable comments.

Reviewer 2 Report

This article don’t necessarily bring new aspects to the scientific literature about this topic, but it is interesting to know the perceptions of health professionals about a new vaccine, particularly at a time when the usefulness of vaccines in the general population is being called into question.

Here are my comments for the authors to help them improve their text.

Abstract:

Line 23: replace “take up” by “Uptake
Line 26-27: there being Cues… in their choice. Line 31-37: No sentence are supported by bibliographic references, references should be indicated.

Line 68: references 24-26, in the text it’s indicated that these references have as population Physician and nurse, out if we look at these references they study other type of populations. Line 86: survey using Convenience sampling. Line 88: Why this age limit between 21-60 years?
Line 95: Employees who agreed to participate in the study signed an informed consent form? Line 146: Table 2 results are presented in one sentence text. This table should be described in detail as the other tables.

It’s the same comment for CI.

Line 221-224: I don’t understand why the adjustment isn’t the same for physicians and nurses, if this is from the results of the univariate analysis, the significant results of the univariate analysis must be presented in a table to see the variables that will be included in the multivariate model.

Line 226-240: This paragraph is complicated to understand as it mixes results from Table 4 and Table 5. It needs to be rewritten to make it more understandable to readers.

Discussion

But it lacks a part "Limit of the study" which is an extremely important part of the discussion.

Line 263: route is not the right term, rephrase.

Line 267: delete Table 3

Author Response

This article don’t necessarily bring new aspects to the scientific literature about this topic, but it is interesting to know the perceptions of health professionals about a new vaccine, particularly at a time when the usefulness of vaccines in the general population is being called into question.

Here are my comments for the authors to help them improve their text.

Q. 1 Abstract:
Line 23: replace “take up” by “Uptake

Authors’ response:

This sentence is reverted to the following under the Abstract section:

A multiple stepwise regression model was used to examine associations between cognitive factors (clinicians’ attitudes) and subjects’ intention to HPV vaccine uptake (L23)

Q.2 Line 26-27: there being Cues… in their choice.

Authors’ response:

The sentence has been reverted to ‘cues to action was featured as a significant background variable in their choice’ (L26-27).

Q.3  L26-27, No sentence are supported by bibliographic references, references should be indicated.

Authors’ response:

Line 26-27 was written the Abstract Section and thus, reference or citations were not required.

Q4. Line 31-37: No sentence are supported by bibliographic references, references should be indicated.

Authors’ response:

References were added back to the main text for Line 31-37 and written as below:

Human papillomavirus (HPV) is a well-known cause of cervical cancer and anogenital warts. HPV-16 and HPV-18 are the most common high-risk HPVs associated with cervical cancer (National Care Institute, 2012); HPV-6 and HPV-11 are lower-risk viruses causing anogenital warts. Genital HPV infections are mainly transmitted by sexual contact (both vaginal and anal) with an infected person (Schiffman et al., 2007). In order to prevent HPV infection, people can either abstain from sex or practise safer sex using condoms. They can cut the risk of infection by having fewer sexual partners. However, epidemiologic studies suggest that up to 75% of all sexually active people will eventually become infected with HPV at some point during their lifetime (Giuliano et al., 2010).

Q.5 Line 68: references 24-26, in the text it’s indicated that these references have as population Physician and nurse, out if we look at these references they study other type of populations. 

Authors’ response:

The references should be Kahn et al (2005) and de Carvalho et al (2009) (reference 24 & 25) in Line 65-67.

Q6. Line 86: survey using Convenience sampling. 

Authors’ response:

We have reverted to “survey using purposive sampling” in Line 85.

Q7. Line 88: Why this age limit between 21-60 years?

Authors’ response:

In Hong Kong, there are four different types of nurses: 1) enrolled nurse, 2) registered nurse, 3) enrolled nurse (psychiatric) and 4) registered nurse (psychiatric). Apart from the Baccalaureate nursing degree which takes a 5-year curriculum, nursing students can choose to take either the two-year diploma or three-year course in the traditional nursing schools to becoming an enrolled or qualified nurse, registered with the Hong Kong Nursing Council. The minimum age of registration is 18 years old. Thus, the youngest age of graduation is usually 20-21 years old. The retirement age in Hong Kong is 60 for nurses and physicians and thus, the age limit is set between 21 and 60 in this study.

Q8. Line 95: Employees who agreed to participate in the study signed an informed consent form?

Authors’ response:

Participants do not require to sign the informed consent form because if they have completed the returned survey to the researchers on-site, it indicated implied consent. Please refer to the sentence in L96-97 as written below:

Returning a completed survey implied consent.

Q9. Line 146: Table 2 results are presented in one sentence text. This table should be described in detail as the other tables.

Authors’ response:

Table 2 has been described in the main text as follows (L147-156)

3.1.  Intention to take up HPV Vaccines (Acceptability),knowledge on HPV vaccines and perceived self-efficacy 

A total of 30.2% of physicians and 21.2% nurses reported an intention to get vaccinated against HPV (Table 2). Less than 40% of physicians (36%, n=64) obtained ≥ five correct responses on knowledge of HPV vaccine compared to 44.4% of nurses (n=436). 70% of physicians and 63% of nurses disagreed that they were not motivated in the HPV vaccine uptake. Rather, 37-41% of nurses and physicians respectively believed that they should take the HPV vaccination. Both physicians and nurses had a relatively high percentage of perceived self-efficacy to complete the full treatment (82% versus 85%) and a significant proportion of physicians and nurses (94% versus 97%) alleged that they were fully autonomous to take up the HPV vaccination if indicated. Physicians and nurses received peer support (76.5% versus 70.5%) and family support to take up the HPV vaccination (78.2% versus 69.3%).
It’s the same comment for CI.

Q.10 Line 221-224: I don’t understand why the adjustment isn’t the same for physicians and nurses, if this is from the results of the univariate analysis, the significant results of the univariate analysis must be presented in a table to see the variables that will be included in the multivariate model.

Authors’ response:

Gender was the only statistically significant background variable among physician participants but not to nurses. Thus, adjustment for female nurses were not required. Only significant variables were thrown into the multiple logistic modelling. Since there were too many tables in the main text already, we thus omitted the univariate analysis table in the text and instead, describe the process in the main text.

Q.11 Line 226-240: This paragraph is complicated to understand as it mixes results from Table 4 and Table 5. It needs to be rewritten to make it more understandable to readers.

Authors’ response:

The paragraph has been rewritten to the following (227-241):

Adjusting for these four variables, results showed that six of the composite cognitive indicator variables were statistically significant. They were knowledge of HPV vaccines (AOR = 2.588, p<0.01), perceived susceptibility to HPV (AOR = 1.779, p<0.05), the perceived benefits of vaccination (AOR = 1.815, p <0.05), perceived self-efficacy (AOR = 2.772, p<0.01), perceived support (AOR = 2.206, p<0.01) and cues to action (AOR = 2.703 to 6.960, p<0.05) (Table 4). In addition, the study fitted a multiple logistic regression model containing all the significant composite cognitive indicator variables and adjusted for significant background variables.

Four background variables were statistically significant among the nurse participants: 1) Gender (ORm = 1.764, 95% CI = 1.129-2.759), 2) age group ("21 - 29": ORm = 4.095, 95% CI = 2.187-7.669; "30 - 39": ORm =1.627,  95% CI = 0.802-3.301, reference group was "40 & over"), 3) having STD-related symptoms (ORm =2.230, 95% CI = 1.779-7.446), and 4) number of sexual acts in the past 6 months ("0": ORm = 1.854, 95% CI = 1.153-2.981; "1 - 5": ORm =2.166, 95% CI = 1.305-3.597, the reference group was "6 or more"). After fitting a multiple logistic regression model containing all the significant cognitive composite indicator variables, and adjusting for the same significant background variable, only perceived self-efficacy and cutes to action emerged as the only significant composite cognitive variables (Table 5).

Q.12 Discussion

But it lacks a part "Limit of the study" which is an extremely important part of the discussion.

Authors’ response:

A section on the Limitations has been added to the main text (L326-331). Please refer to the text below:

Limitations

We used the questions extracted by previous researchers in the HPV studies [17, 30] but these questions had not been validated. There is a potential risk that some of these questions might not accurately measure of what we aimed to measure. Thus, results emerged from this study might not be generalized to similar replicated studies across countries.

Q.13 Line 263: route is not the right term, rephrase.

Authors’ response:

The sentence has been reverted to “their not knowing the risks of acquiring HPV transmission” (L268).

Q.14 Line 267: delete Table 3

Authors’ response:

Table 3 deleted in L267.

Reviewer 3 Report

The study is well designed and reports on an area of high interest in public health.  The results are intriguing and form the basis for future strategies to shape public policy, medical practice and education.  As latent HPV infection poses the potential to drive a variety of cancers this topic addresses a significant issue.

Author Response

The study is well designed and reports on an area of high interest in public health.  The results are intriguing and form the basis for future strategies to shape public policy, medical practice and education.  As latent HPV infection poses the potential to drive a variety of cancers this topic addresses a significant issue.

Authors’ response:

Thank you very much for reviewing our manuscript. Well-appreciated by our authors in this study.

Round 2

Reviewer 2 Report

Thanks to the authors for their corrections and additions to their text.
 I think it’s publishable in the current format